# Synthesis of Poly(Trimethylene Carbonate) from Amine Group Initiation: Role of Urethane Bonds in the Crystallinity

**DOI:** 10.3390/polym13020280

**Published:** 2021-01-16

**Authors:** Thomas Brossier, Gael Volpi, Vincent Lapinte, Sebastien Blanquer

**Affiliations:** 1ICGM, Univ. Montpellier, CNRS, ENSCM, 34296 Montpellier, France; thomas.brossier@3d-medlab.com (T.B.); vincent.lapinte@umontpellier.fr (V.L.); 23D Medlab, 13700 Marignane, France; gael.volpi@3d-medlab.com

**Keywords:** poly(trimethylene carbonate) (PTMC), ring-opening polymerization, amine initiator, urethane

## Abstract

Semi-crystalline poly(trimethylene carbonate) (PTMC) can be efficiently prepared by ring-opening polymerization (ROP) initiated by amine using various catalysts. More promising results were reached with the one-step process of stannous octanoate unlike the two-step one-pot reaction using TBD and MSA catalysts. The ROP-amine of TMC consists in a simple isocyanate free process to produce polycarbonate-urethanes, compatible with the large availability of amines ranging from mono- to multifunctional until natural amino acids. ROP-amine of TMC leads to urethane bonds monitored by FTIR spectroscopy. The relationship between the nature of amines and the crystallinity of PTMC was discussed through X-ray diffraction and thermal studies by DSC and TGA. The impact of the crystallinity was also demonstrated on the mechanical properties of semi-crystalline PTMC in comparison to amorphous PTMC, synthesized by ROP initiated by alcohol. The semi-crystalline PTMC synthesized by ROP-amine opens many perspectives.

## 1. Introduction

Biodegradable synthetic polymers have gained considerable interest in several basic and industrial domains. Among them, aliphatic polycarbonates such as poly(trimethylene carbonate) (PTMC) are remarkable polymers with soft and flexible properties, and present a typical biodegradation mechanism [1]. Indeed, it has been demonstrated that the mechanism of PTMC degradation follows a phenomenon of enzymatic surface erosion without autocatalysis behavior. Hence, the absence of core degradation allows conserving the mechanical properties all along the degradation process, which is a significant difference with their homologous biodegradable aliphatic polyesters [2]. That is why such biodegradability combined with the recognized biocompatibility of the PTMC have yielded especially to high potential in biomedical applications [3]. Hence, PTMC has been used for sustainable and better control drug delivery [4,5] as surface erosion is easier to control than bulk erosion. More particularly, PTMC has been found as material scaffold base in tissue engineering for soft tissue reconstruction, where maintaining the mechanical properties during the degradation phase is crucial [6,7]. Nevertheless, it is globally recognized that the mechanical properties of raw PTMC suffers from several limitations [3,8,9]. Even though the mechanical properties of PTMC evolve with the molar mass, globally, PTMC displays a low modulus combined with a low tensile strength which is often considered as insufficient for many applications. Moreover, PTMC shows low creep resistance which involves poor dimensional stability and is therefore a considerable limitation. Consequently, several chemical approaches have been proposed to conserve the flexibility and the specific biodegradation mechanism while offering mechanical resistance. First of all, chain cohesion reinforcement through PTMC crosslinking has showed positive influence especially in the creep resistance and ultimate stress [10]. Crosslinked PTMC has been obtained either by photo-activation [11] or by ring-opening polymerization of bis-cyclic monomers as linkers [12]. A second approach to improve the mechanical strength was by generating crystalline phase within the amorphous coil of PTMC. Despite few discrepancies around the semi-crystalline properties of PTMC with low molar mass [13], the wide majority of the reported articles present the linear PTMC as a total amorphous polymer in the relaxed state and only semi-crystalline in the stretched state [8,14,15]. Hence, the contribution of crystalline phase within the PTMC material may improve the mechanical strength. In that sense, copolymerization of PTMC with semi-crystalline polymer such as biodegradable polyesters, especially polylactide and poly(ε-caprolactone), have been investigated in block or random copolymerization [9,16] and led to copolymers with the combined homopolymers properties. With the same idea, crystallization of PTMC phase has also been proposed by the formation of self-assembly through the presence of urethane bonds which are widely known to generate hydrogen bonds [17]. So far, poly(carbonate urethane)s have been exclusively studied by classical approach of polyaddition using diisocyanate and polycarbonate terminated diol. PTMC-based polyurethane showed considerable modification of the physical, thermal, and mechanical properties compared to raw PTMC [18,19]. More specifically, the alternation of soft and rigid phases, relayed by the formation of hydrogen bonds, provides excellent elastic properties with large elongation at break and high tensile strength [20,21]. Obviously, the molar mass of the precursor PTMC-terminated diol which represents the soft segments plays an important role in the evolution of the properties.

In this perspective, a more straightforward strategy to generate urethane bonds within the PTMC materials could be by ring opening polymerization (ROP) using aminated initiator, so-called ROP-amine. ROP-amine route has many advantages related to the other pathways of urethanization of PTMC chains. As illustrated in Figure 1, toxic isocyanate reactant is not required and the versatility of amines is much more extended than isocyanates. This variety concerns the functionality of amines (mono, di, tri, multi), the nature of amines (aliphatic, cycloaliphatic, aromatic…), the origin of amines (bio or petrosourced), and so on. Moreover, the strategy is shorter—only one step to reach to poly(carbonate urethane) instead of two steps for post-urethanization. No precaution of stoichiometry is mandatory as the case for polyaddition between diiocyanate and diol precursors. In the literature, ROP-amine has been already investigated for the polymerization of lactide and ε-caprolactone monomers [22,23,24]. ROP-amine for polyesters allows to generate amide groups and shows to modify the properties of the polyester compared to hydroxyl initiation [25]. However, to the best of our knowledge, ROP of TMC has been exclusively initiated with hydroxyl functions but amines have never been studied as initiator for TMC polymerization.

In this study we investigate the capacity of TMC to polymerize by using various amine initiators and describe the effect of the catalyst pathway to induce such ROP-amine. We particularly pay attention to the properties generated by the presence of the urethane bonds on the PTMC.

## 2. Materials and Methods

### 2.1. Materials

Trimethylene carbonate (TMC) comes from Foryou Medical (Huizhou, China). The initiators (1,6-hexanediol, (99%), propylamine, benzylamine, 1,6-hexanediamine, p-xylenediamine, tris(2-aminoethyl)amine, bis(3-aminopropyl)amine, glycine methyl ester, L-phenylalanine methyl ester, the catalysts methanesulfonic acid (MSA), 1,5,7-triazabicyclo[4.4.0]dec-5-ene (TBD), tin(II) 2-ethylhexanoate (Sn(Oct)_2_), were purchased from Sigma Aldrich (Saint-Quentin Fallavier, France) and used as received. Anhydrous dichloromethane was retrieved from solvent purificator Inert PureSolv™ (Castelnau-le-Lez, France) and used as the polymerization solvent.

### 2.2. Methods

#### 2.2.1. Nuclear Magnetic Resonance Spectrometry (NMR)

^1^H NMR and correlated spectroscopy (COSY) NMR spectra were recorded on a 400 MHz and 600 MHz Bruker Aspect Spectrometer. CDCl_3_ was used as deuterated solvent. Chemical shifts were given in parts per million (ppm): for ^1^H NMR the reference peak was residual CDCl_3_ at 7.26 ppm. Degree of polymerization (*DP_n_*) has been calculated with the following equation using protons *H_a_* from the TMC unit (4.2 ppm/integrating for 4 protons), *H_c_* from the initiator (3.1 ppm/integrating for 2x protons, with x the number of arms expected).
(1)DPn=∫Ha4Hc2x

#### 2.2.2. Size Exclusion Chromatograms (SEC)

SEC were recorded using a triple detection (GPC Varian 390-LC viscometer detector, Varian 390-LC refractive index detector and UV detector at 254 nm) from Agilent Technologies (Santa Clara, CA, USA). The analyses were performed in tetrahydrofuran (THF) at a flow rate of 1.0 mL/min at 30 °C. An Agilent PLgel 5 μm guard column and two columns 5 μm PLgel Mixed D were used. Data acquisition and calculations were performed using Cirrus Multi GPC/SEC software. Universal calibration was performed with PS standards from Agilent Technologies using the intrinsic viscosities given by the supplier.

#### 2.2.3. Fourier Transform Infrared Spectra (FT-IR)

FT-IR analyses were recorded on a Perkin Elmer (Villebon-sur-Yvette, France) Spectrum 100 equipped with a ZnSe crystal using the ATR technique. The wave number ranges from 4000 to 650 cm^−1^.

#### 2.2.4. Differential Scanning Calorimetry (DSC)

DSC analyses were carried out using a Mettler Toledo (Mettler Toledo, France,) DSC1 calorimeter. Constant calibration was performed with biphenyl, indium, bismuth, zinc, and cesium chloride standards. Nitrogen was used as the purge gas. Thermal properties were recorded between −50 and 120 °C at a heating rate of 5 °C·min^−1^. Thermal analyses have been initially performed on PTMC_D_-_NH_ after 24 h, 2 weeks, 4 weeks, and 16 weeks (Appendix A). For the rest of the sample all the analyses have been done after 4 weeks at room temperature (20 °C).

#### 2.2.5. Thermogravimetric Analysis (ATG)

The thermal stability was carried out using a NETZSCH TG 209F1 Libra analyzer (Selb, Germany). The experiments consisted in registering the weight loss of the sample under nitrogen flow (40 mL·min^−1^) in function of temperature ranging from 20 up to 600 °C with a heating rate of 10 °C·min^−1^.

#### 2.2.6. Mechanical Measurements

Elongation at break and Young modulus of dumbbell were measured with an Instronb3366L5885 tensile tester (Élancourt, France). Dumbbell were mounted between tensile grips with an initial grip spacing of 10 cm. The cross-head speed was 2 mm·s^−1^. The elongation at break was expressed as a percentage of the original length and the modulus was obtained at 0.5% of deformation by stress/strain. Material film hardness was evaluated with a Shore A durometer (HBA 100-0) from Sauter (Basle, Suisse).

#### 2.2.7. X-Ray Diffraction (XRD)

XRD measurements were performed using a Philips X’Pert PRO MPD apparatus (Amsterdam, The Netherlands) with a copper tube and nickel filter in the range 5 < 2θ < 60°. The measurements were taken every 0.04 rad at room temperature. The XRD patterns were analyzed by PROFIT computer program. The program resolves a diffraction curve on diffraction peaks and amorphous halo which allows estimating the crystallinity degree.

### 2.3. Synthesis

#### 2.3.1. Typical Procedure for Polymerization of TMC Initiated by ROP-amine with Sn(Oct)_2_ (Same Protocol Has Been Performed with All the Amine Initiators) 

Propylamine (1 equivalent), Sn(Oct)_2_ (0.10 wt.%), and trimethylene carbonate (100 equivalent) were charged in a round bottom flask with a stir bar and dried by five successive N_2_/vacuum cycles. The flask was heated to 130 °C using an oil bath. Polymerizations were carried out until the complete conversion of TMC determined by ^1^H NMR. The polymer was dissolved in methylene chloride and collected by precipitation in methanol. The synthesis with 10 equivalent of TMC was also carried out for FT-IR analyses.

^1^H NMR (δ in ppm, CDCl_3_, 400 MHz): 4.7 (m, 1 H, NH), 4.2 (t, 40 or 400 H, CH_2_), 3.8 (m, 2 H, CH_2_), 3.2 (m, 2 H, CH_2_), 2.0 (q, 20 or 200 H, CH_2_), 1.5 (m, 2 H, CH_2_), 0.9 (t, 3 H, CH_3_).

#### 2.3.2. Typical Procedure for Polymerization of TMC Initiated by ROP-Amine with TBD

Propylamine (1 equivalent) was added into a solution of TMC (100 equivalent) in CH_2_Cl_2_ (1.0 mol·L^−1^). The mixture was stirred at 30 °C until the complete consumption of the amine. Then a solution of TBD (0.2 equivalent) in CH_2_Cl_2_ was added to the reaction mixture. After complete reaction, TBD was quenched with benzoic acid. The crude solution was precipitated in methanol.

^1^H NMR (δ in ppm, CDCl_3_, 400 MHz): 4.7 (m, 1 H, NH), 4.2 (t, 40 or 400 H, CH_2_), 3.8 (m, 2 H, CH_2_), 3.2 (m, 2 H, CH_2_), 2.0 (q, 20 or 200 H, CH_2_), 1.5 (m, 2 H, CH_2_), 0.9 (t, 3 H, CH_3_).

## 3. Results and Discussion

### 3.1. Synthesis of PTMC Homopolymer via ROP-Amine Route

In this study, the ring-opening polymerization of TMC initiated by amine groups is presented. The TMC ROP was typically driven by three different catalytic pathways widely used in ROP. To start, we studied the behavior and the efficiency of each catalytic pathway using the amine initiator (Figure 2). A typical degree of polymerization around 100 units has been targeted, and the conversion rate with the molar mass were measured by NMR spectroscopy and chromatography. Catalysis with methanesulfonic acid (MSA) was initially performed. After 24 h of reaction at room temperature, the ^1^H NMR spectroscopy revealed a low conversion rate (around 50%), and the obtained residue did not precipitate in methanol which is usually the case for PTMC (Appendix A). We therefore conclude that the cationic polymerization did not lead to an appropriate polymerization of TMC. Such behavior may be due to the protonation of the amine functions that reduces the nucleophilic character and impairs the ring-opening activation. Consequently, an opposite catalyst with basic properties may be more favorable for initiation of the polymerization. In that sense, we investigated the ROP-amine using the TBD as a nucleophilic catalyst. Interestingly, the polymerization does not show an expected worthy reactivity. In fact, in agreement with recent work obtained with PLA [25], direct anionic polymerization with TBD leads to uncontrolled polymerization of the TMC, with a huge dispersity of 2.8. In addition, we also noticed a significantly longer reactivity with ROP-amine route compared to ROP initiated with hydroxy groups, with respectively only 10% of conversion against 100% after 1 h of reaction [26] (Appendix A). Nevertheless, as proposed by Alba et al. [25], the polymerization can be done in two successive steps one-pot, with first the ring opening of one TMC unit in absence of catalyst, followed by classic ROP initiated by the generated hydroxy groups. By this pre-opening, the polymerization of TMC reached a complete conversion after 1 h with the TBD and 15 h with MSA (Appendix A). However, as the polymerization has been performed in a two-step one-pot approach, this route cannot be considered as real ROP-amine to synthesize the PTMC.

Finally, ROP-amine of TMC has been tested using stannous octanoate catalyst. The ^1^H NMR spectrum from Figure 3a shows the success of the ROP-amine with a conversion close to 100% after 24 h. The conversion rate is calculated by comparing the integrated areas of characteristic peaks of the protons from the monomers at 4.4 ppm with those from the growing polymer chains at 4.2 ppm (*H_a_*). Nevertheless, it is crucial to prove that the generated polymer has been well initiated by 1,6-hexanediamine. For that, 2D COSY NMR (Figure 3b) spectroscopy allowed to correlate a new peak at 3.1 ppm (*H_c_*) to the characteristic peak of the amino group at 4.7 ppm (*H_f_*), and also correlated to the characteristic peak of the protons from the aliphatic 1,6-hexanediamine at 1.5 ppm (*H_d_*) which can be assigned to the protons in alpha position of the resulting urethane. This peak has been used to calculate the average degree of polymerization in number of the PTMC which was close to the expected one (DP_100_) (Table 1).

Once the formation of PTMC was demonstrated, the reactivity of TMC using 1,6-hexanediamine was compared to a traditional initiation with 1,6-hexanediol. Figure 4a represents the kinetic analysis (monomer conversion versus time) of ROP by both initiators using Sn(Oct)_2_ catalyst. Interestingly, no differences have been noticed regarding the reactivity profile of each polymerization demonstrating a similar kinetic polymerization. For both initiations, the polymerization reached a plateau at 90% of conversion after 10 h of reaction. Moreover, in Figure 4b with both initiators, the PTMC samples were characterized by a similar monodisperse SEC trace with a close molar mass and dispersity (1.8 and 1.9 respectively) as already described for the synthesis of PLA and PCL [23].

The strategy of ROP-amine of TMC has been investigated through a series of aminated initiators (PTMC_NH_) related to alcohol initiator (PTMC_OH_) (Table 1). The study of aminated initiators extends from monoamines (primary aliphatic (PTMC_M_-_NH_), aromatic (PTMC_M_-_NH_-_Φ_), and secondary aliphatic (PTMC_M_-_S_-_NH_)) to diamines (primary aliphatic (PTMC_D_-_NH_) and aromatic (PTMC_D_-_NH_-_Φ_), and multifunctional amines (PTMC_T_-_NH_). The study was pursued with protected amino esters of glycine (PTMC_M_-_AAG_-_NH_) and phenylalanine (PTMC_M_-_AAP_-_NH_). Such study demonstrates the versatility of this approach to generate multiurethane functions along the PTMC chains.

From Table 1, it appears that the *DP_n_* determined by NMR spectroscopy and chromatography are both consistent and similar with the theoretical targeted molar mass. The dispersities determined by SEC ranged from 1.6 to 1.9, except for PTMC_T_-_NH_ which is significantly higher around 3.4 and can be explained by its branched structure. As it can be expected, the mono and multifunctional aminated initiators induce a same efficiency of polymerization of the PTMC. Interestingly, secondary amine showed also efficient initiation of the polymerization, as it was also demonstrated for ROP-amine of lactide and lactone [27]. This is interesting for deeper investigation and comparison with primary amine, because secondary amine generates substituted urethane without hydrogen bonds. Following the study of Liu et al. on the polymerization of the PCL [28], we successfully synthesize the PTMC by ROP-amine using natural amino acid initiators. In order to reduce eventual by-reactions during the ROP-amine, the amino acids were protected into amino methylic esters. Such natural amino acid models are a huge potential to open a new possibility to use biosourced initiators for PTMC polymerization. The different molar masses with the theoretical ones can be explained by a partial activation of the initiator due to unfavored interactions or steric hindrance of the initiator provoking higher molar masses than theoretically expected.

### 3.2. Urethane Bond Formation Determined by Infra-Red Analyses

The urethane groups along the PTMC chains resulting from the initiation step of the ROP-amine of TMC was monitored by FTIR spectroscopy using various amine and diamine initiators in reference to PTMC_D_-_OH_ (Figure 5). The study was realized on short PTMC chains (DP_10_) to magnify the urethane contribution related to carbonate ones. The characteristic carbonate signals of υ(C=O) and υ(C–O–C) around 1741 and 1222 cm^−1^, respectively were detected for all polymers including PTMC_D_-_OH_. The comparison between mono-, di-, and triamine initiators put in relieve the number of urethane groups along the PTMC chains with the increase of the amide II band intensity (δ(N–H) + υ(C–N)) at 1523 cm^−1^. Interestingly, we clearly see the apparition of supplementary peak around 1700 cm^−1^ for PTMC_D_-_NH_ et PTMC_T_-_NH_ but totally absent for PTMC_OH_. As described in the literature this peak corresponds to the carbonyl involved in the H-bonds (named H-bonds υ(C=O) in the Figure 5) [29]. This can explain the increased trend in function of the number of urethane groups.

### 3.3. Thermal and Mechanical Properties of PTMC from ROP-Amine

A thermal study was realized by thermogravimetric analysis to evaluate the thermal stability of PTMC bearing urethane groups as illustrated in Table 2 and Figure 6a. PTMC_OH_ and PTMC_NH_ have the same thermal degradation profile without any weight loss before 250 °C and a sharp one-step weight loss around 280 °C. No notable difference was detected with the number of urethane groups per chain because of the low number of urethane groups related to the carbonate units. Consequently, the traditional characteristic urethane degradation in three stages was not observed [30]. Hence, this route of urethanization avoids the problematic release of HCN at 420–460 °C, inherent to the degradation of urethane groups, which gives a supplementary advantage.

A thermal study of PTMC was also monitored by DSC to estimate the amorphous or semi-crystalline character of the polymers. Such study allows also to determine the potential influence in the crystalline organization generated by the presence of H-bonds as shown in FTIR (Figure 5). Indeed, poly(carbonate urethane) synthesized via the traditional urethanization with isocyanate led to semi-crystalline PTMC with evolved crystallinity in function of the hard segment content [18]. It appears that the crystallization kinetics was rather slow (Appendix A) and no crystallization has been found for any samples after 24 h. In addition, thermal analysis after 4 and 16 weeks did not show significant differences, we therefore performed all the characterizations after 4 weeks. The DSC thermograms are presented in Figure 6b. All PTMC_NH_ have a low glass transition temperature around −20/−27 °C related to PTMC_D_-_OH_ (−20 °C in accordance to literature [8]) (Table 2). We noted the amorphous character of PTMC initiated by alcohol (PTMC_D_-_OH_) while PTMC initiated by amine (excepted to PTMC_M_-_NH_) were semi-crystalline with a melting temperature around 40 °C. A noteworthy melting phenomenon appears for the PTMC initiated by amine, excepted for PTMC_M_-_NH_. The urethane groups in PTMC chains promote the ordered polymers. For instance, for linear PTMC, the crystallinity increased with the urethane groups per chain starting from amorphous character without urethane unit (PTMC_D_-_OH_) or with one (PTMC_M_-_NH_) to semi-crystalline character for two urethane groups and still more with four urethane obtained by classical urethanization approach using propylisocyanate as described in Appendix A. In addition, we noted that PTMC_D_-_NH_-_Φ_ (Δ*H_m_* = 49.4 J·g^−1^) showed a similar crystallization with this from the PTMC_D_-_NH_ (Δ*H_m_* = 52.5 J·g^−1^). Therefore, this resemblance shows that the aromatic stacking did not influence or increase the crystallization behavior as it could have been expected. As illustrated with tri-arms star-shaped (PTMC_T_-_NH_) using tris(2-aminoethyl)amine, the crystallinity of PTMC decreased with the number of arms regarding the enthalpy of PTMC_D_-_NH_ and PTMC_T_-_NH_ (Δ*H_m_* = 52.5 and 37.3 J·g^−1^, respectively). Therefore, the branching of the PTMC structure yield to a different chain organization compared to the linear PTMC_NH_ which then disadvantages the crystallization.

The impact of urethane bonds on thermal properties was also assessed for higher molar mass using a PTMC with *DP_n_* around 200 and 500. Surprisingly, as summarized in Table 3, a similar crystallinity was measured for all the *DP_n_* with a melting temperature around 41 °C, and a similar melting enthalpy around 50 J·g^−1^. Such semi-crystalline character for high molar mass of PTMC has never been reported before. Hence, we can hypothesize that the hydrogen bonds from the urethane groups organize the polymer chains and therefore favor the crystallization of the PTMC chains (Figure 7). It has to be noted that despite the low ratio of urethane groups, the crystallization occurs regardless the molar mass.

These results demonstrate the impact of urethane groups along the PTMC chains on the organization of the polymer chains and induce the crystallization phenomenon by hydrogen bonds. This trend shows the efficiency of this strategy of urethanization by ROP-amine instead of post-urethanization of PTMC_OH_. Other consequences of the urethanes on PTMC were shown using mechanical tests. First, the molding samples of PTMC_D_-_NH_ contrasted with the amorphous PTMC_D_-_OH_ material, in the rubbery state at room temperature, too soft to be molded. This different behavior has been measured by hardness shore A with 15 ± 1.5 for PTMC_D_-_OH_ and 82 ± 1.2 for PTMC_D_-_NH_. Young Modulus of PTMC_D_-_NH_ with DP_100_ was also determined by tensile test (E = 36.2 MPa, σ_max_ = 0.95 MPa) and comforted the significant stiff properties. In opposition, PTMC_D_-_OH_ using the same molar mass was not measurable because of the huge creep behavior which was not the case for PTMC_D_-_NH_. Nevertheless, due to the crystallinity, PTMC_D_-_NH_ remains quite fragile (ε_max_ = 2%) and larger improvement could be obtained by cross-linking PTMC_D_-_NH_ to reinforce the mechanical properties as it is traditionally proposed in the literature for PTMC_OH_ [8,10].

### 3.4. X-Ray Diffraction Analyses

The crystallization character of ROP-amine PTMC was finally investigated by XRD analyses. The analyses were performed on the samples that showed the highest crystalline ratio in DSC, including PTMC_D_-_NH_-_Φ_ and PTMC_D_-_NH_ for *DP_n_* at 100, 200, and 500 (Figure 8). For all the samples, four characteristic peaks of crystalline PTMC_NH_ are observed with the highest double peak 2θ around 21–22°, a sharper peak at 16°, and an additional small peak at 27°. Consequently, the presence of such sharp peaks in XRD proves the existence in the polymer samples of a relatively well-ordered phase which attests the occurrence of a crystalline phase. The X-ray diffraction profile of PTMC_NH_ closely resembled to the semi-crystalline poly(alkylene carbonate) with high number of CH_2_ groups [31], or else the PTMC after crystallization upon stretching [8]. Indeed, XRD pattern of these poly(alkylene carbonate) specimen are characterized by strong reflections between 20 and 22.5° and by weaker reflections between 25 and 25.8°. In addition, diffraction peaks exhibited a similar location for all the measured samples, suggesting that the crystalline phase was similar for the samples and present a similar proportion of crystalline phase. However, it has to be noticed that the characteristic sharp peak that described the crystalline phase do not return to the base line which is typically due to the overlapping with the wide signal traditionally obtained with amorphous phases. This behavior is in agreement with the literature like in the case of the copolymer of PLLA-PTMC, which shows the apparition of crystalline peaks characteristic to PLLA crystallization which overlap the wide halo due to the amorphous phase of PTMC [32]. The degree of crystallinity was determined for all the tested PTMC_NH_ around 33%. The XRD results were consistent with the DSC data and can then give another proof that the assembled crystalline phase was generated by the formation of hydrogen bonds from the urethane groups.

### 3.5. Interest of Semi-Crystalline PTMC

To conclude, it can be noticed that the ROP-amine approach drastically modifies the properties of the raw PTMC traditionally obtained by the alcohol initiation. The capacity to melt the PTMC can lead to considerable benefit in the processability, especially where amorphous PTMC cannot be directly applied such as in extrusion process or additive manufacturing process like 3D-FDM [33]. Moreover, the urethane bonds for the primary amine initiators have proved to generate hydrogen bonds and therefore may lead to the formation of self-assembled objects which can be promising for biomedical applications especially for the end-capped urethane obtained by amino acid initiators [25]. Furthermore, generated multiurethane bonds in the middle of the PTMC chain, using multifunctionalized amine initiators, can be used as precursors in polyaddition for the synthesis of new poly(carbonate urethane)s in order to maximize the number of urethane functions.

## 4. Conclusions

Herein, the first ROP-amine of trimethylene carbonate was successfully achieved and compared to ROP initiated by alcohol. The results of polymerization with Sn(Oct)_2_ were better than those with TBD and MSA catalysts and similar to an alcohol initiation in terms of kinetic of polymerization and dispersity of molar mass. The efficiency of ROP-amine to produce poly(carbonate urethane)s was extended to a large platform of petro- and biobased amines. Many poly(carbonate urethane)s have semi-crystalline character even for high molar mass (*DP_500_*), until now always described as amorphous, showing the importance of hydrogen bonds in crystallization process. This easy free-isocyanate route to semi-crystalline poly(carbonate urethane)s opens many perspectives to hybrid materials using amino acids or toward new industrial processes such as extrusion.

## Figures and Tables

**Figure 1 polymers-13-00280-f001:**
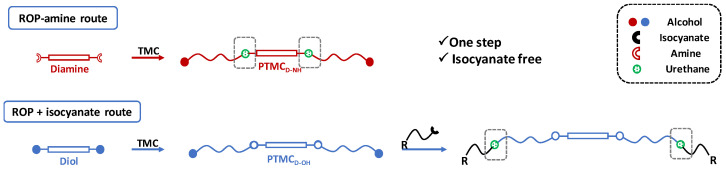
Comparative routes toward poly(trimethylene carbonate) (PTMC)-urethane by ring-opening polymerization (ROP)-amine route and polyaddition of isocyanate in PTMC terminal diols.

**Figure 2 polymers-13-00280-f002:**
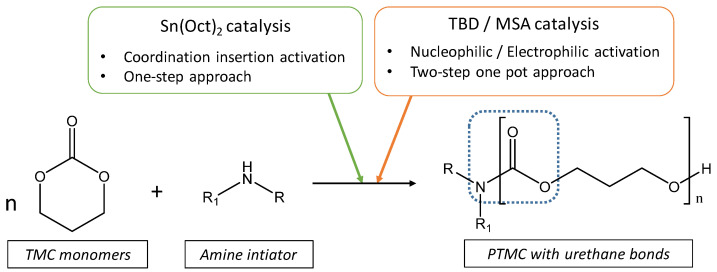
Overview on the catalysis pathways for the ROP-amine to synthesize the PTMC.

**Figure 3 polymers-13-00280-f003:**
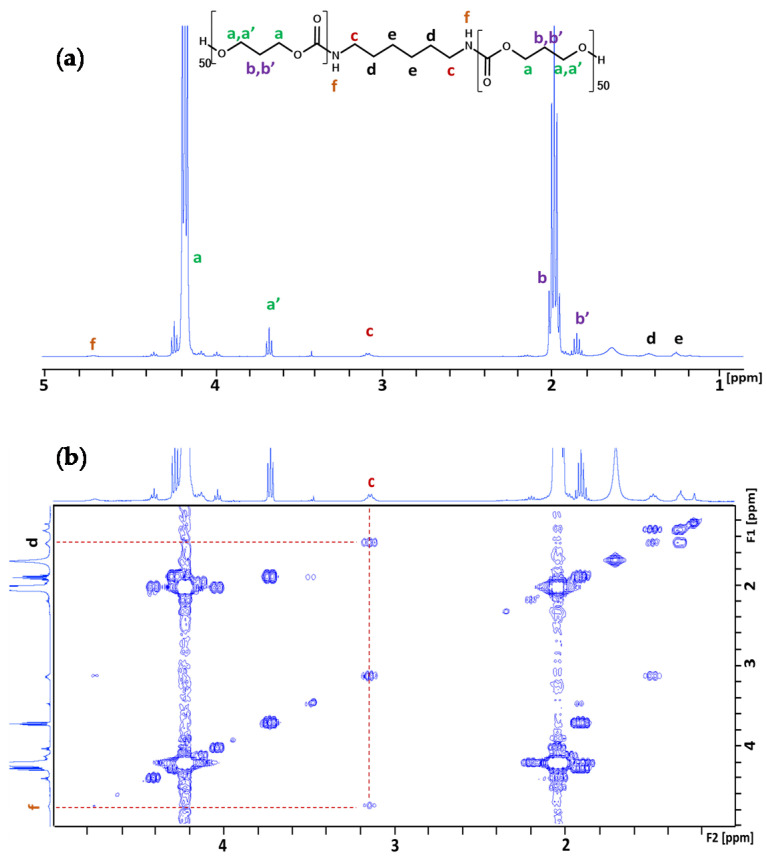
(**a**) ^1^H NMR and (**b**) 2D-COSY NMR of PTMC initiated by 1,6-hexanediamine with Sn(Oct)_2_ (CDCl_3_).

**Figure 4 polymers-13-00280-f004:**
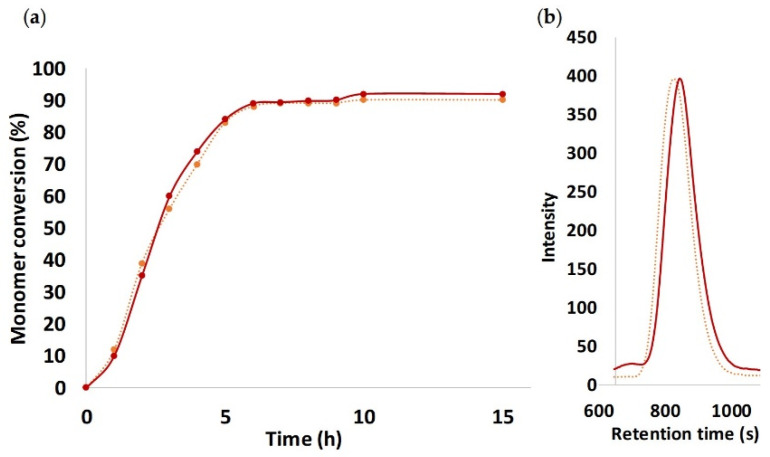
Comparison of (**a**) kinetic profile of PTMC initiated by 1,6-hexanediamine using Sn(Oct)_2_ (line red) 1,6-hexanediol (dots orange), and (**b**) SEC traces of the resulting polymers.

**Figure 5 polymers-13-00280-f005:**
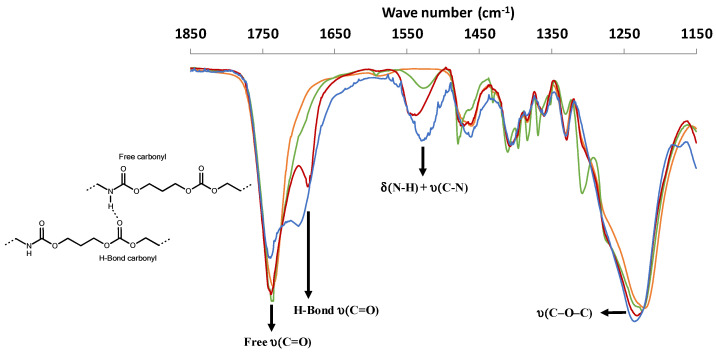
FTIR spectra of PTMC_D_-_OH_ (orange), PTMC_M_-_NH_ (green), PTMC_D_-_NH_ (red), and PTMC_T_-_NH_ (blue).

**Figure 6 polymers-13-00280-f006:**
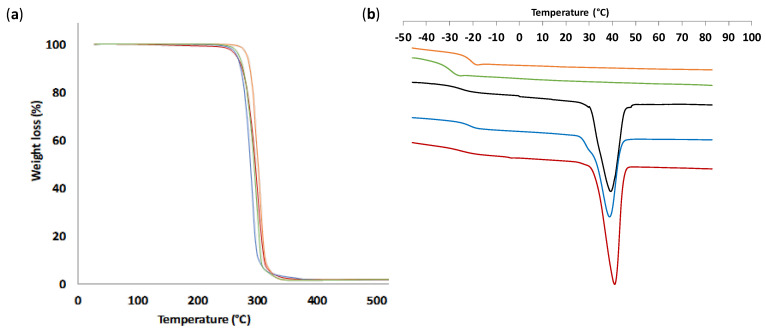
Thermal properties of PTMC_D_-_OH_ (orange); PTMC_M_-_NH_ (green); PTMC_T_-_NH_ (blue); PTMC_D_-_NH_ (red); PTMC_D_-_NH_-_Φ_ (black). (**a**) Thermal gravimetric analysis; (**b**) differential scanning calorimetry traces.

**Figure 7 polymers-13-00280-f007:**
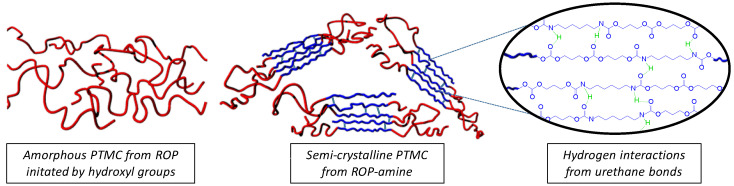
Illustration of amorphous and semi-crystalline organization of PTMC chains. Red worms represent the amorphous phase while blue worms illustrate the crystalline phase.

**Figure 8 polymers-13-00280-f008:**
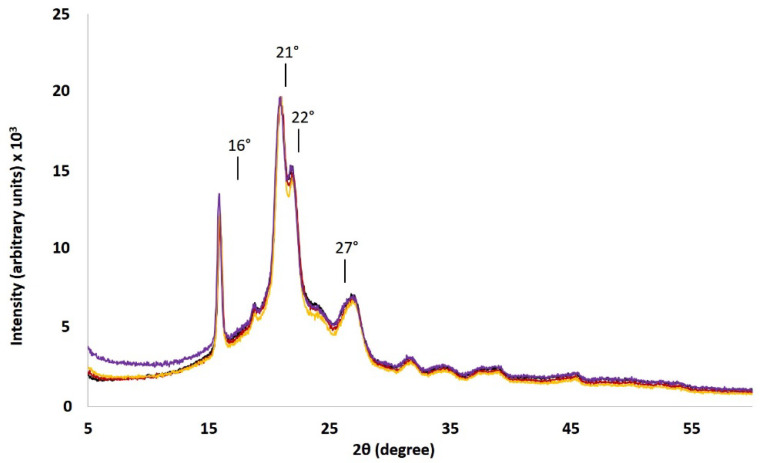
XRD patterns of the PTMC_D_-_NH_-_Φ_ (black) and PTMC_D_-_NH_ with *DP_n_* at 100 (red), 200 (yellow) and 500 (purple).

**Table 1 polymers-13-00280-t001:** Molar mass of PTMC_OH_ and various PTMC_NH_ for *DP_n_* = 100.

Initiator Formula	Initiator Name	Abbreviation of PTMC	*F* ^1^	*DP_n_* _NMR_	*DP_n_* _SEC_	*Ð* ^3^
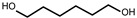	1,6-hexanediol	PTMC_D_-_OH_	D ^2^	93	110	1.8
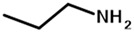	propylamine	PTMC_M_-_NH_	M ^2^	88	86	1.9
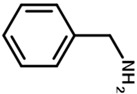	benzylamine	PTMC_M_-_NH_-_Φ_	M	88	86	1.9
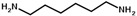	1,6-hexanediamine	PTMC_D_-_NH_	D	102	86	1.9
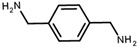	*p*-xylenediamine	PTMC_D_-_NH_-_Φ_	D	108	98	1.7
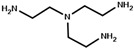	tris (2-aminoethyl)amine	PTMC_T_-_NH_	T ^2^	98	83	3.4
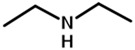	bis(3-aminopropyl)amine	PTMC_M_-_S_-_NH_	M_s_ ^2^	94	108	1.6
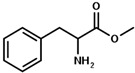	L-phenylalanine methyl ester	PTMC_M_-_AAP_-_NH_	M	143	72	1.6
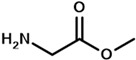	glycine methyl ester	PTMC_M_-_AAG_-_NH_	M	147	70	1.7

^1^ Functionality; ^2^ M: monofunctional; D: difunctional; T: trifunctional; Ms: substituted monofunctional; ^3^ Dispersity.

**Table 2 polymers-13-00280-t002:** Thermal properties of various alcohol and amine initiated PTMC (*DP_n_* = 100).

PTMC	T_g_ ^1^ (^°^C)	T_m_ ^1^ (^°^C)	∆*H_m_*^1^ (J·g^−1^)	T_10%_ ^2^ (^°^C)
PTMC_D_-_OH_	−20.0	-	-	288
PTMC_M_-_NH_	−30.0	-	-	276
PTMC_D_-_NH_	−25.0	40.8	52.5	275
PTMC_D_-_NH_-_Φ_	−24.6	39.2	49.4	276
PTMC_T_-_NH_	−22.3	38.7	37.3	273

^1^ values determined by DSC; ^2^ values determined by TGA after 10% weight loss.

**Table 3 polymers-13-00280-t003:** Investigation of high molar mass PTMC.

PTMC	*DP_n_* _th_	*DP_n_* _NMR_ ^1^	T_g_ ^2^ (^°^C)	T_m_ ^2^ (^°^C)	∆*H_m_*^2^ (J·g^−1^)
PTMC_D_-_NH_	100	102	−25.0	40.8	52.5
PTMC_D_-_NH_	200	196	−27.8	41.9	51.7
PTMC_D_-_NH_	500	460	−26.4	40.7	48.8

^1^ NMR 600 MHz; ^2^ values determined by DSC.

## Data Availability

Data sharing not applicable.

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
