# Peer review of "Synthesis of Poly(Trimethylene Carbonate) from Amine Group Initiation: Role of Urethane Bonds in the Crystallinity"

_polymers, 2021, doi:10.3390/polym13020280_

Round 1

Reviewer 1 Report

The paper describes the use of amines as initiators for the ROP of TMC. The interesting and original information reported in this paper relies on the preparation of semi-crystalline polyTMC instead of classically amorphous polyTMC when alcohols are used as initiators.

The section devoted to the synthesis is quite long because, even though the data are new, they are not unexpected, based on similar studies carried out for the ROP of lactones (the corresponding papers are cited). For instance, it is expected that acids protonate amines, which are no more nucleophilic with a detrimental effect on ROP. A short note mentioning that TBD and MSA were tested but did not allowed to achieve the controlled synthesis (data can be provided in supporting information). The description of the results obtained with Sn(Oct)2 is enough.

The characterization of the mechanical properties is important because this is the original part of the paper, but this section is too short. It seems to me that it is important to characterize the crystallinity of the same samples used for the characterization of mechanical properties to prove a correlation between mechanical properties and crystallinity. For a same architecture, does crystallinity change with the number of urethane groups (1 or 2 depending on a mono-amine or a di-amine is used as an initiator)? Besides, the effect of branching could also be analyzed and discussed as far as star-shaped polyTMC are concerned (use of at least -tri-functional tri-amines). The effect of the presence of urethane groups on crystallization kinetics is also interesting to analyze and is not commented in the paper. Is crystallization fast or slow?

Below, you will find a listing of remarks, which are obviously major.

Nevertheless, I believe that after update of the paper, it could be original enough to deserve publication. I recommend thus to publish this paper after a major revision.

Remarks

  • Please use “molar mass” instead of “molecular weight” according to the more recent IUPAC recommendations
  • Lign 172: ROP catalyzed or initiated by any species through activation through H-bonds is not anionic, cationic and is also typical and not mentioned in the sentence.
  • Lign 171: The sentence “ROP of cyclic monomers is typically driven by three different catalytic mechanisms such as cationic, anionic and coordination-insertion” is dangerous because no detailed study of the mechanism is reported. It is more cautious to report that ROP was achieved by using 3 typical catalysts/ initiators. This sentence is always true whatever the mechanism.
  • In relation to my last comment, is ROP catalyzed by MSA purely cationic? If this is the case, the catalytic activity should depend on pKa and trifluoromethanesulfonic should be more efficient regarding polymerization kinetics. It is reported, at least for the ROP of lactones, that mechanism is more complicated as assumed by a purely cationic mechanism. Again, the authors could be more cautious regarding the mechanism and they can just slightly change the text to avoid to enter in that discussion, which is beyond the scope of their paper.
  • Lign 187: “anionic polymerization with TBD”. Again, several mechanisms are reported, and it is not proved in this paper that anionic species are involved in the mechanism. The anionic mechanism implies the presence of an amide or an alkoxide. An alcohol or an amine activated through H-bonds is more nucleophilic but is not an anion when the heteroatom-H bond is not fully broken. Again, the authors should be more cautious.
  • Lign 195: not clear because ROP initiated by amines will always be in 2-steps: initiation by the amine and propagation through the alcohol at the chain-end. I follow that the authors favored a one-pot approach rather than a 2-pot approach. The pending question on the 2 steps are based on their respective kinetics (are they simultaneous or not?). Please clarify.
  • Figure 3 : it is impossible to read the too small chemical shift on the figure
  • 8 Line 210: The calculation of molar mass by NMR is fine but the detailed equation used for the calculation should be clearly written.
  • Line 245: I do not understand why the authors compared molar masses measured by NMR and SEC (Table 1). The values measured by NMR are real values whereas values measured by SEC are not the real values due to the use of the refraction index detector and a PS calibration.
  • Lign 286: did the authors achieved the analysis of the gases released during thermal degradation to claim that no HCL was released? A direct proof is required!
  • Line 294 and Table 2: Is the DSC thermogram recoded during the first of the second scan. The paper is not clear, but it seems to be the first scan with the risk that crystallinity change from sample to sample due to a different thermal history. To erase such an effect, DSC reported in Table 2 should be recorded during the second scan and this should be clearly mentioned in the paper.
  • Line 318: The authors claim “hydrogen bondings result not only from urethane bonds themselves but also between urethane and carbonate bonds” I don’t follow how the authors can draw this conclusion just based on the high molar mass of the PTMC they prepared. Is there a direct proof supporting this claim? This is a key point in the paper because it is reported that these H-bonds have an impact on crystallization.

Reviewer 2 Report

Comments:

This is a very interesting work about versatile synthesis of poly(trimethylene carbonate) (PTMC) from amine group initiation and role of urethane bonds on the properties. It provides a new,  efficient  and simple isocyanate free synthesis approach and detailed discussion on the relationship between the nature of amines and the crystallinity of PTMC prepared. This could be of interest to broad researchers. Based on this, I suggest accept as is.

However, I would suggest the authors reconsider the manuscript title because the important findings  and information are missing (especially the second part). 

Reviewer 3 Report

The article presents the results of studies on synthesis of poly(trimethylene carbonate) from amine group initiation. In order to verify the assumptions and characterize materials, the Authors use research methods adequate to the issues considered. The article is interesting and presents issues related to new polymer materials. The title fully reflects the content contained in the article, and the summary is written correctly. The methodology is presented correctly and contains information that allows the reproduction of research by other researchers.

Before publishing the article, however, I have some comments and questions on the reviewed work:

  • Were DSC tests performed as one heating run only?
  • Table 2 - some of the results are given with the decimal point, and some not.
  • The mechanical properties should be further described. There is no information about the conditions and parameters of tensile tests.

Reviewer 4 Report

I have found the work very interesting and well done, thus, I suggest its acceptance after minor revision.

My only remark is about the chain size obtained with amino acids in table 1, because DPn determined by NMR and SEC are very different (around 140 and 70 respectively), and it may deserve some commentary. It seems that the ester functionality is interfering with the polymerization process forming some kind of by-products, thus distorting the integral of the NMR signals, and consuming around the 30% of the TMC (what could explain the DPn determined by chromatography). A suggestion for future work could be the optimization of the reaction conditions to improve this result, if it's worth doing.

Some typographic mistakes:

Line 235: PTMCD-NH, D-NH in subindex

Line 270: PTMCT-NH, T-NH in subindex

Line 281: Tableau 2 should be Table 2

Round 2

Reviewer 1 Report

The authors revised their paper according to my comments. They also give satisfactory answers to my questions. Some pending points remain but they are minor, and I recommend thus the paper for publication.

Remarks

  • As far as 1,6-hexanediamine is the initiator as shown in figure 3, the area under peak a is divided by 4 ( 2 CH2 groups), which is OK, but the area under the peak c should be divided by 4 and not by 2 to take into account the 2 CH2-N-CO groups. The problem is that the initiator is not clearly written in the text and the calculation can slightly change depending on the initiator even though the principle remains the same. Please clarify

  • The authors revised their paper by mentioning “it appears that the DPn determined by NMR spectroscopy and chromatography are both consistent”. I have the feeling that my comment was not crystal clear. I agree that both techniques are important and complementary. GPC allows to measure the dispersity and the profile of the distribution curve of number of chains and the apparent molar mass (in other words, the chromatogram). Both techniques give a different information because the real molar mass is only given by NMR but not by GPC. I do not see any criteria to decide the consistency of the molar mass (or DPn) found by NMR and GPC with a PS calibration unless a detector giving the real molar mass is used (light scattering detector) for GPC.

  • Regarding the question on TGA, the released gas is of course HCN and not HCL. Sorry for this typo mistake. I follow the answer of the authors even though it is an indirect proof. It remains interesting to achieve the characterization of the released gases, harvested at the outlet of the TGA, by mass spectrometry or IR to highlight the mechanism of the degradation but I agree that this analysis goes beyond the scope of this first paper.
